# Unifying and Boosting Gradient-Based Training-Free Neural Architecture Search

**Yao Shu**[†]**, Zhongxiang Dai**[†]**, Zhaoxuan Wu**[§]**, Bryan Kian Hsiang Low**[†]

Dept. of Computer Science, National University of Singapore, Republic of Singapore[†]
Institute of Data Science, National University of Singapore, Republic of Singapore[§]
Integrative Sciences and Engineering Programme, NUSGS, Republic of Singapore[§]
{shuyao,daizhongxiang,lowkh}@comp.nus.edu.sg[†]
wu.zhaoxuan@u.nus.edu[§]

## Abstract

*Neural architecture search* (NAS) has gained immense popularity owing to its ability to automate neural architecture design. A number of training-free metrics are recently proposed to realize NAS without training, hence making NAS more scalable. Despite their competitive empirical performances, a unified theoretical understanding of these training-free metrics is lacking. As a consequence, *(a)* the relationships among these metrics are unclear, *(b)* there is no theoretical interpretation for their empirical performances, and *(c)* there may exist untapped potential in existing training-free NAS, which probably can be unveiled through a unified theoretical understanding. To this end, this paper presents a unified theoretical analysis of gradient-based training-free NAS, which allows us to *(a)* theoretically study their relationships, *(b)* theoretically guarantee their generalization performances, and *(c)* exploit our unified theoretical understanding to develop a novel framework named *hybrid NAS* (HNAS) which consistently boosts training-free NAS in a principled way. Remarkably, HNAS can enjoy the advantages of both training-free (i.e., the superior search efficiency) and training-based (i.e., the remarkable search effectiveness) NAS, which we have demonstrated through extensive experiments. Our codes are available at https://github.com/shuyao95/HNAS.

## 1 Introduction

Recent years have witnessed a surging interest in applying *deep neural networks* (DNNs) in real-world applications, e.g., machine translation [1], object detection [2], among others. To achieve compelling performances in these applications, many domain-specific neural architectures have been handcrafted by human experts with considerable efforts. However, these efforts have gradually become unaffordable due to the growing demand for customizing neural architectures for different tasks. To this end, *neural architecture search* (NAS) [3] has been proposed to design neural architectures automatically. While many *training-based* NAS algorithms [4, 5] have achieved state-of-the-art (SOTA) performances in various tasks, their search costs usually are unaffordable in resource-constrained scenarios mainly due to their requirement for training DNNs during search. As a result, a number of training-free metrics have been developed to realize *training-free NAS* [6, 7]. Surprisingly, these training-free NAS algorithms are able to achieve competitive empirical performances even compared with other training-based NAS algorithms while incurring significantly reduced search costs. Moreover, the architectures selected by these training-free NAS algorithms have been empirically found to transfer well to different tasks [7, 8].

---

Correspondence to: Zhongxiang Dai <daizhongxiang@comp.nus.edu.sg>.

36th Conference on Neural Information Processing Systems (NeurIPS 2022).

Despite the impressive empirical performances of the NAS algorithms using training-free metrics, *a unified theoretical analysis of these training-free metrics* is still lacking in the literature, leading to a few significant implications. Firstly, the theoretical relationships of these training-free metrics are unclear, making it challenging to explain *why they usually lead to comparable empirical results* [9]. Secondly, there is no theoretical guarantee for the empirically observed compelling performances of the architectures selected by NAS algorithms using these training-free metrics. As a consequence, the reason *why NAS using these training-free metrics works well* is still not well understood, and hence there lacks theoretical assurances for NAS practitioners when deploying these algorithms. To the best of our knowledge, the theoretical aspect of NAS with training-free metrics has only been preliminarily studied by Shu et al. [8]. However, their analyses are only based on the training rather than generalization performances of different architectures and are restricted to a *single* training-free metric. Thirdly, there may exist untapped potential in existing training-free NAS algorithms, which probably can be unveiled through a unified theoretical understanding of their training-free metrics.

To this end, we perform a unified theoretical analysis of *gradient-based* training-free NAS to resolve all the three problems discussed above in this paper. Firstly, we theoretically prove the connections among different gradient-based training-free metrics in Sec. 4.1. Secondly, based on these provable connections, we derive a unified generalization bound for DNNs with these metrics and then use it to provide principled interpretations for the compelling empirical performances of existing training-free NAS algorithms (Secs. 4.2 and 4.3). Moreover, we demonstrate that our theoretical interpretation for training-free NAS algorithms, surprisingly, displays the same preference of architecture topology (i.e., wide or deep) as training-based NAS algorithms under certain conditions (Sec. 4.4), which helps to justify the practicality of our theoretical interpretations. Thirdly, by exploiting our unified theoretical analysis, we develop a novel NAS framework named *hybrid NAS* (HNAS) to consistently boost existing training-free NAS algorithms (Sec. 5) in a principled way. Remarkably, through a theory-inspired combination with *Bayesian optimization* (BO), our HNAS framework enjoys the advantages of both training-based (i.e., remarkable search effectiveness) and training-free (i.e., superior search efficiency) NAS simultaneously, making it more advanced than existing training-free and training-based NAS algorithms. Lastly, we use extensive experiments to verify the insights derived from our unified theoretical analysis, as well as the search effectiveness and efficiency of our non-trivial HNAS framework (Sec. 6).

## 2   Related Works

Recently, a number of training-free metrics have been proposed to estimate the generalization performances of neural architectures, allowing the model training in NAS to be completely avoided. For instance, Mellor et al. [6] have developed a heuristic metric using the correlation of activations in an initialized DNN. Meanwhile, Abdelfattah et al. [9] have empirically revealed a large correlation between training-free metrics that were formerly applied in network pruning, e.g., SNIP [10] and GraSP [11], and the generalization performances of candidate architectures in the search space. These results hence indicate the feasibility of using training-free metrics to estimate the performances of candidate architectures in NAS. Chen et al. [7] have proposed a heuristic metric to trade off the trainability and expressibility of neural architectures in order to find well-performing architectures in various NAS benchmarks. Xu et al. [12] have applied the mean of the Gram matrix of gradients to quickly estimate the performances of architectures. More recently, Shu et al. [8] have employed the theory of *Neural Tangent Kernel* (NTK) [13] to formally derive a performance estimator using the trace norm of the NTK matrix with initialized model parameters, which, surprisingly, is shown to be data- and label-agnostic. Though these existing works have demonstrated the feasibility of training-free NAS through their compelling empirical results, the reason as to *why training-free NAS performs well in practice* and the answer to the question of *how training-free NAS can be further boosted* remain mysteries in the literature. This paper aims to provide theoretically grounded answers to these two questions through a unified analysis of existing gradient-based training-free metrics.

## 3   Notations and Backgrounds

### 3.1   Neural Tangent Kernel

To simplify the analysis in this paper, we consider a $L$-layer DNN with identical widths $n_1 = \cdots = n_{L-1} = n$ and scalar output (i.e., $n_L = 1$) based on the formulation of DNNs in [13]. Let $f(\boldsymbol{x}, \boldsymbol{\theta})$

be the output of a DNN with input $\boldsymbol{x} \in \mathbb{R}^{n_0}$ and parameters $\boldsymbol{\theta} \in \mathbb{R}^d$ that are initialized using the standard normal distribution, the NTK matrix $\boldsymbol{\Theta} \in \mathbb{R}^{m \times m}$ over a dataset of size $m$ is defined as

$$\boldsymbol{\Theta}(\boldsymbol{x}, \boldsymbol{x}'; \boldsymbol{\theta}) = \nabla_{\boldsymbol{\theta}} f(\boldsymbol{x}, \boldsymbol{\theta})^\top \nabla_{\boldsymbol{\theta}} f(\boldsymbol{x}', \boldsymbol{\theta}) . \tag{1}$$

Jacot et al. [13] have shown that this NTK matrix $\boldsymbol{\Theta}$ will finally converge to a deterministic form $\boldsymbol{\Theta}_\infty$ in the infinitely wide DNN model. Meanwhile, Arora et al. [14], Allen-Zhu et al. [15] have further proven that a similar result, i.e., $\boldsymbol{\Theta} \approx \boldsymbol{\Theta}_\infty$, can also be achieved in over-parameterized DNNs of finite width. Besides, Arora et al. [14], Lee et al. [16] have revealed that the training dynamics of DNNs can be well-characterized using this NTK matrix at initialization (i.e., $\boldsymbol{\Theta}_0$ based on the initialized model parameters $\boldsymbol{\theta}_0$) under certain conditions. More recently, Yang and Littwin [17] have further demonstrated that these conclusions about NTK matrix shall also hold for DNNs with any reasonable architecture, even including recurrent neural networks (RNNs) and graph neural networks (GNNs). Therefore, the conclusions drawn based on the formulation above in this paper are expected to be applicable to the NAS search spaces with complex architectures, which we will validate empirically.

### 3.2 Gradient-Based Training-Free Metrics for NAS

In this paper, we mainly focus on the study of those *gradient-based* training-free metrics, i.e., the training-free metrics that are derived from the gradients of initialized model parameters, which we introduce below. Previous works have empirically shown that better model performances are usually associated with larger values of these training-free metrics in practice [9].

**Gradient norm of initialized model parameters.**   While Abdelfattah et al. [9] were the first to employ the gradient norm of initialized model parameters to estimate the generalization performance of candidate architectures, the same form has also been derived by Shu et al. [8] to approximate their training-free metric efficiently. Following the notations in Sec. 3.1, let $\ell(\cdot, \cdot)$ be the loss function, we define the gradient norm over dataset $S = \{(\boldsymbol{x}_i, y_i)\}_{i=1}^m$ as

$$\mathcal{M}_{\mathrm{Grad}} \triangleq \left\| \frac{1}{m} \sum_{i=1}^m \nabla_{\boldsymbol{\theta}} \ell(f(\boldsymbol{x}_i, \boldsymbol{\theta}_0), y_i) \right\|_2 . \tag{2}$$

**SNIP and GraSP.**   SNIP [10] and GraSP [11] were originally proposed for training-free network pruning, and Abdelfattah et al. [9] have applied them in training-free NAS to estimate the performances of candidate architectures without model training. Following the notations in Sec. 3.1, let $\mathbf{H}_i \in \mathbb{R}^{d \times d}$ denote the hessian matrix induced by input $\boldsymbol{x}_i$, the metrics of SNIP and GraSP on dataset $S = \{(\boldsymbol{x}_i, y_i)\}_{i=1}^m$ can be defined as

$$\mathcal{M}_{\mathrm{SNIP}} \triangleq \left| \frac{1}{m} \sum_i^m \boldsymbol{\theta}_0^\top \nabla_{\boldsymbol{\theta}} \ell(f(\boldsymbol{x}_i, \boldsymbol{\theta}_0), y_i) \right| , \quad \mathcal{M}_{\mathrm{GraSP}} \triangleq \left| \frac{1}{m} \sum_i^m \boldsymbol{\theta}_0^\top \left( \mathbf{H}_i \nabla_{\boldsymbol{\theta}} \ell(f(\boldsymbol{x}_i, \boldsymbol{\theta}_0), y_i) \right) \right| . \tag{3}$$

Of note, we use the scaled (i.e, by $1/m$) absolute value of the original GraSP metric in [11] throughout this paper to match the mathematical form of other training-free metrics.

**Trace norm of NTK matrix at initialization.**   Recently, Shu et al. [8] have reformulated NAS into a constrained optimization problem to maximize the trace norm of the NTK matrix at initialization. In addition, Shu et al. [8] have empirically shown that this trace norm is highly correlated with the generalization performance of candidate architectures under their derived constraint. Let $\boldsymbol{\Theta}_0$ be the NTK matrix based on initialized model parameters $\boldsymbol{\theta}_0$ of a DNN, then without considering the constraint in [8], we frame this training-free metric on dataset $S = \{(\boldsymbol{x}_i, y_i)\}_{i=1}^m$ as

$$\mathcal{M}_{\mathrm{Trace}} \triangleq \sqrt{\|\boldsymbol{\Theta}_0\|_{\mathrm{tr}} / m} . \tag{4}$$

## 4   Theoretical Analyses of Training-Free NAS

### 4.1   Connections among Training-Free Metrics

Notably, though the gradient-based training-free metrics introduced in Sec. 3.2 seem to have distinct mathematical forms, most of them will actually achieve similar empirical performances in practice [9].

More interestingly, these metrics in fact share the similarity of using the gradients of initialized model parameters in their calculations. Based on these facts, we propose the following hypothesis to explain the similar performances achieved by different training-free metrics in Sec. 3.2: *The training-free metrics in Sec. 3.2 may be theoretically connected and hence could provide similar characterization for the generalization performances of neural architectures*. We validate this hypothesis affirmatively and use the following theorem to establish the theoretical connections among these metrics.

**Theorem 1.** *Let the loss function $\ell(\cdot, \cdot)$ in gradient-based training-free metrics be $\beta$-Lipschitz continuous and $\gamma$-Lipschitz smooth in the first argument. There exist the constant $C_1, C_2, C_3 > 0$ such that the following holds with a high probability,*

$$\mathcal{M}_{\text{Grad}} \leq C_1 \mathcal{M}_{\text{Trace}}, \ \mathcal{M}_{\text{SNIP}} \leq C_2 \mathcal{M}_{\text{Trace}}, \ \mathcal{M}_{\text{GraSP}} \leq C_3 \mathcal{M}_{\text{Trace}} \ .$$

The proof of Theorem 1 are given in Appendix A.1. Notably, our Theorem 1 implies that with a high probability, architectures of larger $\mathcal{M}_{\text{Grad}}$, $\mathcal{M}_{\text{SNIP}}$ or $\mathcal{M}_{\text{Grad}}$ will also achieve a larger $\mathcal{M}_{\text{Trace}}$ given the inequalities above. That is, the value of $\mathcal{M}_{\text{Grad}}$, $\mathcal{M}_{\text{SNIP}}$ and $\mathcal{M}_{\text{Grad}}$ for different architectures in the NAS search space should be highly correlated with the value of $\mathcal{M}_{\text{Trace}}$. As a consequence, these training-free metrics should be able to provide similar estimation of the generalization performances of architectures (validated in Sec. 6.2) and hence similar performances can be achieved when using these metrics (validated in Sec. 6.4). Overall, the training-free NAS metrics from Sec. 3.2 can all be theoretically connected with $\mathcal{M}_{\text{Trace}}$ despite their distinct mathematical forms. Note that though our Theorem 1 is only able to establish the theoretical connections between $\mathcal{M}_{\text{Trace}}$ and other training-free metrics, our empirical results in Appendix C.1 further reveal that *any two* training-free metrics from Sec. 3.2 will also be highly correlated. Interestingly, these results also serve as principled justifications for the similar performances achieved by these training-free metrics in [9].

## 4.2 A Generalization Bound Induced by Training-free Metrics

Let dataset $S = \{(\boldsymbol{x}_i, y_i)\}_{i=1}^m$ be randomly sampled from a data distribution $\mathcal{D}$, we denote $\mathcal{L}_S(\cdot)$ as the training error on $S$ and $\mathcal{L}_\mathcal{D}(\cdot)$ as the corresponding *generalization error* on $\mathcal{D}$. Intuitively, a smaller generalization error indicates a better *generalization performance*. Thanks to the common theoretical underpinnings of gradient-based training-free metrics formalized by Theorem 1, we can perform a *unified generalization analysis* for DNNs in terms of these metrics by making use of the NTK theory [13]. Define $\ell(f, y) \triangleq (f - y)^2/2$ and $\eta_0 \triangleq \min\{2n^{-1}(\lambda_{\min}(\boldsymbol{\Theta}_\infty) + \lambda_{\max}(\boldsymbol{\Theta}_\infty))^{-1}, m\lambda_{\max}^{-1}(\boldsymbol{\Theta}_0)\}$ with $\lambda_{\min}(\cdot), \lambda_{\max}(\cdot)$ being the minimum and maximum eigenvalue of a matrix respectively, we derive the following theorem:

**Theorem 2.** *Assume $\|\boldsymbol{x}_i\|_2 \leq 1$ and $f(\boldsymbol{x}_i, \boldsymbol{\theta}_0), \lambda_{\min}(\boldsymbol{\Theta}_0), y_i \in [0, 1]$ for any $(\boldsymbol{x}_i, y_i) \in S$. There exists a constant $N \in \mathbb{N}$ such that for any $n > N$, when applying gradient descent with learning rate $\eta < \eta_0$, the generalization error of $f_t$ at time $t > 0$ can be bounded as below with a high probability,*

$$\mathcal{L}_\mathcal{D}(f_t) \leq \mathcal{L}_S(f_t) + \mathcal{O}(\kappa/\mathcal{M}) \ .$$

*Here, $\mathcal{M}$ can be any metric in Sec. 3.2 and $\kappa \triangleq \lambda_{\max}(\boldsymbol{\Theta}_0)/\lambda_{\min}(\boldsymbol{\Theta}_0)$ is the condition number of $\boldsymbol{\Theta}_0$.*

Its proof is in Appendix A.2 and the second term $\mathcal{O}(\kappa/\mathcal{M})$ in Theorem 2 represents the *generalization gap* of DNN models. Notably, our Theorem 2 provides an explicit theoretical connection between the gradient-based training-free metrics from Sec. 3.2 and the generalization gap of DNNs, which later serves as the foundation to theoretically interpret the compelling performances achieved by existing training-free NAS algorithms (Sec. 4.3). Compared to the traditional Rademacher complexity [18], these training-free metrics provide alternative methods to measure the complexity of DNNs when estimating the generalization gap of DNNs.

## 4.3 Concrete Generalization Guarantees for Training-Free NAS

Since the $\mathcal{L}_S(\cdot)$ in our Theorem 2 may also depend on the training-free metric $\mathcal{M}$, it also needs to be taken into account when analyzing the generalization performance (or the generalization error $\mathcal{L}_\mathcal{D}(\cdot)$) for training-free NAS methods. To this end, in this section, we derive concrete generalization guarantees for NAS methods using training-free metrics by considering two different scenarios (i.e., the *realizable* and *non-realizable* scenarios) for the training error term $\mathcal{L}_S(\cdot)$ in Theorem 2, which finally give rise to principled interpretations for different training-free NAS methods [7–9].

**The realizable scenario.** Similar to [18], we assume that a zero training error (i.e., $\mathcal{L}_S(f_t) \to 0$ when $t$ is sufficiently large) can be achieved in the realizable scenario. By further assuming that the condition number $\kappa$ in Theorem 2 is bounded by $\kappa_0$ for all candidate architectures in the search space, we can then derive the following generalization guarantee (Corollary 1) for the realizable scenario.

**Corollary 1.** *Under the conditions in Theorem 2, for $f_t$ at convergence (i.e., $t \to \infty$) in the realizable scenario and for any training-free metric $\mathcal{M}$ from Sec. 3.2, the following holds with a high probability,*

$$\mathcal{L}_{\mathcal{D}}(f_t) \leq \mathcal{O}(1/\mathcal{M}) .$$

Corollary 1 is obtained by introducing $\mathcal{L}_S(f_t) = 0$ and $\kappa \leq \kappa_0$ into Theorem 2. Importantly, Corollary 1 suggests that in the realizable scenario, the generalization error of DNNs is negatively correlated with the metrics from Sec. 3.2. That is, an architecture with a larger value of training-free metric $\mathcal{M}$ generally achieves an improved generalization performance. This implies that in order to select well-performing architectures, we can simply maximize $\mathcal{M}$ to find $\mathcal{A}^* = \arg\max_{\mathcal{A}} \mathcal{M}(\mathcal{A})$ where $\mathcal{A}$ denotes any architecture in the search space. Interestingly, this formulation aligns with the training-free NAS method from [9], which has made use of the metrics $\mathcal{M}_{\text{Grad}}$, $\mathcal{M}_{\text{SNIP}}$ and $\mathcal{M}_{\text{GraSP}}$ to achieve good empirical performances. Therefore, our Corollary 1 provides a valid generalization guarantee and also a principled justification for the method from [9].

**The non-realizable scenario.** In practice, different candidate architectures in a NAS search space typically have diverse non-zero training errors [8] and $\kappa$ [7]. Therefore, the assumptions of the zero training error and the bounded $\kappa$ in the realizable scenario above may be impractical. In light of this, we drop these two assumptions and derive the following generalization guarantee (Corollary 2) for the non-realizable scenario, which, interestingly, facilitates theoretically grounded interpretations for the training-free NAS methods from [8, 7].

**Corollary 2.** *Under the conditions in Theorem 2, for any $f_t$ at time $t > 0$ and any training-free metric $\mathcal{M}$ from Sec. 3.2 in the non-realizable scenario, there exists a constant $C > 0$ such that with a high probability,*

$$\mathcal{L}_{\mathcal{D}}(f_t) \leq \frac{1}{2} \left( m - \eta \mathcal{M}^2/C \right)^{2t} + \mathcal{O}(\kappa/\mathcal{M}) .$$

Its proof is given in Appendix A.3. Notably, our Corollary 2 suggests that when $\mathcal{M} \in [0, \sqrt{mC/\eta}]$, an architecture with a larger value of the metric $\mathcal{M}$ will lead to a better generalization performance because such a model has both a faster convergence (i.e., the first term decreases faster w.r.t time $t$) and a smaller generalization gap (i.e., the second term is smaller). Interestingly, Shu et al. [8] have leveraged this insight to introduce the training-free metric of $\mathcal{M}_{\text{Trace}}$ with a constraint, which has achieved a higher correlation with the generalization performance of architectures than the metrics from [9]. This therefore implies that our Corollary 2 followed by [8] provides a better characterization of the generalization performance of architectures than Corollary 1 followed by [9] since the non-realizable scenario we have considered will be more realistic than the realizable scenario as explained above. Meanwhile, Corollary 2 also suggests that there exists a trade-off in terms of $\mathcal{M}$ between the model convergence (i.e., the first term) and the generalization gap (i.e., the second term) when $\mathcal{M} > \sqrt{mC/\eta}$, which surprisingly is similar to the empirically motivated trainability and expressivity trade-off in [7]. In addition, Corollary 2 also indicates that for architectures achieving similar values of $\mathcal{M}$, the ones with smaller condition numbers $\kappa$ generally achieve better generalization performance. Interestingly, such a result also aligns with the conclusion from [7]. Therefore, our Corollary 2 also provides a principled justification for the training-free NAS method in [7].

### 4.4 Connection to Architecture Topology

Interestingly, we can prove that the condition number $\kappa$ in our Corollary 2 is theoretically related to the architecture topology, i.e., whether the architecture is wide (and shallow) or deep (and narrow), to further support the practicality and the superiority of our Corollary 2. In particular, inspired by the theoretical analysis from [19], we firstly analyze the eigenvalues of the NTK matrices of two different architecture topologies (i.e., wide vs. deep architectures), which gives us an insight into the difference between their corresponding $\kappa$. We mainly consider the following wide (i.e., $f$) and deep (i.e., $f'$) architecture illustrated in Figure 3, respectively:

$$f(\boldsymbol{x}) = \mathbf{1}^{\top} \sum_{i=1}^{L} \mathbf{W}^{(i)} \boldsymbol{x} , \; f'(\boldsymbol{x}) = \mathbf{1}^{\top} (\prod_{i=1}^{L} \mathbf{W}^{(i)}) \boldsymbol{x} \qquad (5)$$

where $\mathbf{W}^{(i)} \in \mathbb{R}^{n \times n}$ for any $i \in \{1, \cdots, L\}$ and every element of $\mathbf{W}^{(i)}$ is independently initialized using the standard normal distribution. Here, $\mathbf{1}$ denotes an $n$-dimensional vector with every element being one. Let $\mathbf{\Theta}_0$ and $\mathbf{\Theta}_0'$ be the NTK matrices of $f$ and $f'$ that are evaluated on the finite dataset $S = \{(\boldsymbol{x}_i, y_i)\}_{i=1}^m$, respectively, we derive the following theorem:

**Theorem 3.** *Let dataset $S$ be normalized using its statistical mean and covariance such that $\mathbb{E}[\boldsymbol{x}] = 0$ and $\mathbf{X}^\top \mathbf{X} = \mathbf{I}$ given $\mathbf{X} \triangleq [\boldsymbol{x}_1 \boldsymbol{x}_2 \cdots \boldsymbol{x}_m]$, we have*

$$\mathbf{\Theta}_0 = Ln \cdot \mathbf{I} \,, \ \mathbb{E}\left[\mathbf{\Theta}_0'\right] = Ln^L \cdot \mathbf{I} \,.$$

Its proof is in Appendix A.4. Notably, Theorem 3 shows that the NTK matrix of the wide architecture in (5) is guaranteed to be a scaled identity matrix, whereas the NTK matrix of the deep architecture in (5) is a scaled identity matrix *only in expectation* over random initialization. Consequently, we always have $\kappa = 1$ for the initialized wide architecture, while $\kappa > 1$ with high probability for the initialized deep architecture. Also note that as we have discussed in Sec. 4.3, our Corollary 2 shows that (given similar values of $\mathcal{M}$) an architecture with a smaller $\kappa$ is likely to generalize better. Therefore, this implies that wide architectures generally achieve better generalization performance than deep architectures (given similar values of $\mathcal{M}$). This, surprisingly, aligns with the findings from [19] which shows that wide architectures are preferred in *training-based NAS* due to their competitive performances in practice, thus further implying that our Corollary 2 is more practical and superior to our Corollary 1. More interestingly, based on the definition of $\mathcal{M}_{\text{Trace}}$ (4), Theorem 3 also indicates that deep architectures are expected to have larger values of $\mathcal{M}_{\text{Trace}}$ (due to the larger scale of $\mathbb{E}\left[\mathbf{\Theta}_0'\right]$ for deep architectures) and hence achieve larger model complexities than wide architectures.

## 5 Hybrid Neural Architecture Search

### 5.1 A Unified Objective for Training-Free NAS

Our theoretical understanding of training-free NAS in Sec. 4 finally allows us to address the following question in a principled way: *How can we consistently boost existing training-free NAS algorithms?* Specifically, to realize this target, we propose to select well-performing architectures by minimizing the upper bound on the generalization error in Corollary 2 given any training-free metric from Sec. 3.2. We expect this choice to lead to improved performances over the method from [9] because Corollary 2 provides a more practical generalization guarantee for training-free NAS than Corollary 1 followed by [9] (Sec. 4.3). Formally, let $\mathcal{A}$ be any architecture in the search space and let $\mathcal{M}$ be any training-free metric from Sec. 3.2, then NAS problem can be formulated below in a unified manner:

$$\min_{\mathcal{A}} \frac{1}{2} \left(m - \eta \mathcal{M}^2(\mathcal{A})/C\right)^{2t} + \mathcal{O}\left(\kappa(\mathcal{A})/\mathcal{M}(\mathcal{A})\right) \,. \tag{6}$$

We further reformulate (6) into the following form:

$$\min_{\mathcal{A}} \kappa(\mathcal{A})/\mathcal{M}(\mathcal{A}) + \mu F(\mathcal{M}^2(\mathcal{A}) - \nu) \tag{7}$$

where $F(x) \triangleq x^{2t}$, and $\mu$ and $\nu$ are hyperparameters we introduced to absorb the impact of all other parameters in (6). Compared with the diverse form of NAS objectives in [7–9], our (7) presents a non-trivial unified form of NAS objectives for all the training-free metrics from Sec. 3.2, making it easier for practitioners to deploy NAS with different types of evaluated training-free metrics. Our NAS objective in (7) is a natural consequence of our generalization guarantee in Corollary 2 and therefore will be more theoretically grounded, in contrast to the heuristic objective in [7]. Moreover, our (7) advances the training-free NAS method based on $\mathcal{M}_{\text{Trace}}$ from [8], because our (7) *(a)* is derived from the generalization error instead of the training error (that is followed by [8]) of DNNs, which therefore will be more sound and practical, *(b)* have unified all the gradient-based training-free metrics from Sec. 3.2, and *(c)* have considered the impact of condition number $\kappa$ which is shown to be critical in practice (see our Appendix C.2). Above all, our unified NAS objective in (7) is expected to be able to lead to improved performances over other existing training-free NAS methods.

### 5.2 Optimization and Search Algorithm

Our theoretically motivated NAS objective in (7) has unified all training-free metrics from Sec. 3.2 and improved over existing training-free NAS methods. However, its practical deployment requires

---

**Algorithm 1:** Hybrid Neural Architecture Search (HNAS)

---

1: **Input:** Training and validation dataset, metric $\mathcal{M}$ evaluated on architecture pool $\mathcal{P}$, $F(\cdot)$ for (7),
   evaluation history $\mathcal{H}_0 = \varnothing$, a BO algorithm $\mathcal{B}$, number of iterations/queries $K$
2: **for** iteration $k = 1, \ldots, K$ **do**
3:     Choose $\mu_k, \nu_k$ using the BO algorithm $\mathcal{B}$
4:     Obtain the optimal candidate $\mathcal{A}_k^*$ in $\mathcal{P}$ by solving (7)
5:     Evaluate the validation performance of $\mathcal{A}_k^*$, e.g., $\mathcal{L}_{\text{val}}(\mathcal{A}_k^*)$ after training $\mathcal{A}_k^*$
6:     Update the GP surrogate in the BO algorithm $\mathcal{B}$ using the evaluation history
       $\mathcal{H}_k = \mathcal{H}_{k-1} \bigcup \{((\mu_k, \nu_k), \mathcal{L}_{\text{val}}(\mathcal{A}_k^*))\}$
7: **end for**
8: Select the final $\mathcal{A}^*$ with the best validation performance, e.g., $\mathcal{A}^* = \arg\min_{\mathcal{A} \in \{\mathcal{A}_k^*\}_{k=1}^K} \mathcal{L}_{\text{val}}(\mathcal{A})$

---

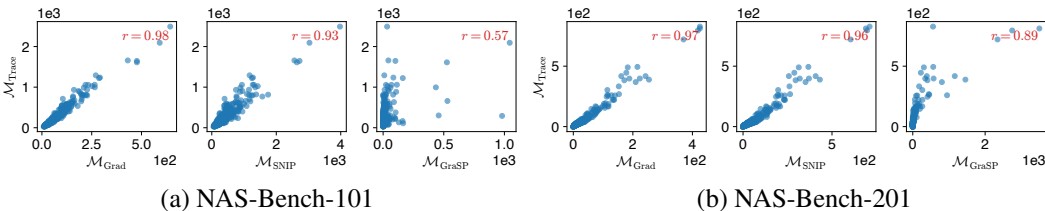

Figure 1: Spearman correlation between $\mathcal{M}_{\text{Trace}}$ and other training-free metrics from Sec. 3.2, which are evaluated in NAS-Bench-101/201. The correlation coefficient $r$ is given in the corner of each plot.

the determination of the hyperparameters $\mu$ and $\nu$,[1] which can be non-trivial in practice. To this end, we further introduce *Bayesian optimization* (BO) [20] to optimize the hyperparameters $\mu$ and $\nu$ in order *to maximize the true validation performance* of the architectures selected by different $\mu$ and $\nu$. In particular, BO uses a Gaussian process (GP) as a surrogate to model the objective function (i.e., the validation performance here) in order to sequentially choose the queried inputs (i.e., the values of $\mu$ and $\nu$). This finally completes our theoretically grounded NAS framework called *hybrid NAS* (HNAS), which not only novelly unifies all training-free metrics from Sec. 3.2 but also boosts NAS algorithms based on these training-free metrics in a principled way (Algorithm 1).

Specifically, in every iteration $k$ of HNAS, we firstly select the optimal candidate $\mathcal{A}_k^*$ by maximizing our training-free NAS objective in (7) using the values of $\mu$ and $\nu$ queried by the BO algorithm in the current iteration (line 3-4 of Algorithm 1). Next, we evaluate the validation performance of $\mathcal{A}_k^*$ (e.g., validation error $\mathcal{L}_{\text{val}}(\mathcal{A}_k^*)$) and then use it to update the GP surrogate that is applied in the BO algorithm (line 5-6 of Algorithm 1), which then will be used to choose the values of $\mu$ and $\nu$ in the next iteration. After HNAS completes, the final selected architecture is chosen as the one achieving the best validation performance among all the optimal candidates, e.g., $\mathcal{A}^* = \arg\min_{\mathcal{A} \in \{\mathcal{A}_k^*\}_{k=1}^K} \mathcal{L}_{\text{val}}(\mathcal{A})$ (see Appendix B.1 for more optimization details of Algorithm 1). Thanks to the utilization of validation performance as the objective for BO, our HNAS is expected to be able to enjoy the advantages of both training-free (i.e., the superior search efficiency) and training-based NAS (i.e., the remarkable search effectiveness) as supported by our extensive empirical results in Sec. 6.4. In addition, by novelly introducing BO to optimize the low-dimensional continuous hyperparameters $\mu$ and $\nu$ rather than the high-dimensional discrete architectural hyperparameters in the NAS search space, HNAS is able to avoid the issues of high-dimensional discrete optimization that standard BO algorithms usually attain when they are directly applied for NAS [21], allowing HNAS to be more efficient and effective in practice as empirically supported in our Sec. 6.4.

## 6 Experiments

### 6.1 Connections among Training-Free Metrics

We firstly validate the theoretical connections between $\mathcal{M}_{\text{Trace}}$ and other training-free metrics from Sec. 3.2 by examining their Spearman correlations for all architectures in NAS-Bench-101 [22] and

---

[1]Of note, we usually fix $t = 1$, which is already reasonably good for $F(\cdot)$. So, the practical deployment of (7) will mainly be affected by the choice of $\mu$ and $\nu$.

Table 1: Correlation coefficients between the test errors evaluated on CIFAR-10 and the generalization guarantees in Sec. 4.3 for the architectures in NAS-Bench-101/201.

| Metric | NAS-Bench-101 | | NAS-Bench-201 | |
|---|---|---|---|---|
| | Spearman | Kendall's Tau | Spearman | Kendall's Tau |
| **Realizable scenario** | | | | |
| $\mathcal{M}_{\text{Grad}}$ | −0.25 | −0.17 | 0.64 | 0.47 |
| $\mathcal{M}_{\text{SNIP}}$ | −0.21 | −0.15 | 0.64 | 0.47 |
| $\mathcal{M}_{\text{GraSP}}$ | −0.45 | −0.31 | 0.57 | 0.40 |
| $\mathcal{M}_{\text{Trace}}$ | −0.30 | −0.21 | 0.54 | 0.39 |
| **Non-realizable scenario** | | | | |
| $\mathcal{M}_{\text{Grad}}$ | 0.35 | 0.23 | 0.75 | 0.56 |
| $\mathcal{M}_{\text{SNIP}}$ | 0.37 | 0.25 | 0.75 | 0.56 |
| $\mathcal{M}_{\text{GraSP}}$ | 0.46 | 0.32 | 0.69 | 0.50 |
| $\mathcal{M}_{\text{Trace}}$ | 0.33 | 0.23 | 0.70 | 0.51 |

Table 2: Comparison of topology, $\mathcal{M}_{\text{Trace}}$ & $\kappa$ of different architectures. The topology width/depth of each architecture is followed by the maximum value in the search space (separated by a slash).

| Architecture | Topology | | $\mathcal{M}_{\text{Trace}}$ | $\kappa$ |
|---|---|---|---|---|
| | Width | Depth | | |
| NASNet | 5.0/5.0 | 2/6 | 31±2 | 118±41 |
| AmoebaNet | 4.0/5.0 | 4/6 | 36±2 | 110±39 |
| ENAS | 5.0/5.0 | 2/6 | 36±2 | 98±33 |
| DARTS | 3.5/4.0 | 3/5 | 33±2 | 122±58 |
| SNAS | 4.0/4.0 | 2/5 | 31±2 | 126±47 |
| WIDE | 4.0/4.0 | 2/5 | 27±1 | 141±36 |
| DEEP | **1.5**/4.0 | **5**/5 | **131±16** | **209±107** |

NAS-Bench-201 [23] with CIFAR-10 [24]. Figure 1 illustrates the result where all these training-free metrics are evaluated using a batch (with size 64) of sampled data following that of [9]. Of note, we will follow the same approach to evaluate these training-free metrics in our following sections. The results in Figure 1 show that $\mathcal{M}_{\text{Trace}}$ and other training-free metrics from Sec. 3.2 are indeed highly correlated since they consistently achieve high positive correlations in different search spaces. These empirical results actually align with the interpretation of our Theorem 1 (Sec. 4.1). Moreover, the correlation between any two training-free metrics from Sec. 3.2 is in Appendix C.1, which further verifies the connection among all these training-free metrics. Above all, in addition to the theoretical justification in our Theorem 1, our empirical results have also supported the connections among all the training-free metrics from Sec. 3.2.

## 6.2 Generalization Guarantees for Training-Free NAS

We then demonstrate the validity of our generalization guarantees for training-free NAS (Sec. 4.3) by examining the correlation between the generalization bound in the realizable (Corollary 1) or non-realizable (Corollary 2) scenario and the test errors of architectures in NAS-Bench-101/201. Similar to HNAS (Algorithm 1), we employ BO with a sufficiently large number of iterations (e.g., hundreds of iterations) to determine the non-trivial parameters in Corollary 2. Table 1 summarizes the results on CIFAR-10 where a higher positive correlation implies a better agreement between our generalization guarantee and the generalization performance of architectures. Notably, the generalization bound in the realizable scenario performs a compelling characterization of the test errors in NAS-Bench-201 with relatively high positive correlations, whereas it fails to provide a precise characterization in a larger search space, i.e., NAS-Bench-101. Remarkably, our generalization bound in the non-realizable scenario is able to perform consistent improvement over it by obtaining higher positive correlations. These results imply that the Corollary 1 may only provide a good characterization for training-free NAS in certain cases (e.g., in the small-scale search space NAS-Bench-201), whereas our Corollary 2 generally is more valid and robust in practice. As a consequence, our (6) following Corollary 2 should be able to improve over the NAS objective following Corollary 1 as we have justified in Sec. 5. Interestingly, the comparable results achieved by all training-free metrics from Sec. 3.2 again validate the connections among these metrics (Theorem 1). Moreover, our additional results in Appendix C.2 further confirm the validity and practicality of our generalization guarantees for training-free NAS.

## 6.3 Connection to Architecture Topology

To support the theoretical connections between architecture topology (wide vs. deep) and the value of training-free metric $\mathcal{M}_{\text{Trace}}$ as well as the condition number $\kappa$ shown in Sec. 4.4, we compare the topology width/depth, $\mathcal{M}_{\text{Trace}}$ and $\kappa$ of the architectures selected by different SOTA training-based NAS algorithms in the DARTS search space, including NASNet [25], AmoebaNet [26], ENAS [4], DARTS [5], and SNAS [27]. Table 2 summarizes the results where we apply the same definition of topology width/depth in [19] (refer to [19] for more details). We also include the widest (called WIDE) and the deepest (called DEEP) architecture in the DARTS search space into this comparison. As shown in our Table 2, wide architectures (i.e., all architectures except DEEP) consistently achieve lower condition number $\kappa$ and smaller values of $\mathcal{M}_{\text{Trace}}$ than deep architecture (i.e., DEEP), which aligns with our theoretical insights in Sec. 4.4.

Table 3: Comparison of NAS algorithms in NAS-Bench-201. The result of HNAS is reported with the mean and standard deviation of 5 independent searches and its search costs are evaluated on a Nvidia 1080Ti. C & D in the last column denote continuous and discrete search space, respectively.

| Algorithm | Test Accuracy (%) | | | Cost (GPU Sec.) | Method | Applicable Space |
|---|---|---|---|---|---|---|
| | C10 | C100 | IN-16 | | | |
| ResNet [28] | 93.97 | 70.86 | 43.63 | - | manual | - |
| REA[†] | 93.92±0.30 | 71.84±0.99 | 45.15±0.89 | 12000 | evolution | C & D |
| RS (w/o sharing)[†] | 93.70±0.36 | 71.04±1.07 | 44.57±1.25 | 12000 | random | C & D |
| REINFORCE[†] | 93.85±0.37 | 71.71±1.09 | 45.24±1.18 | 12000 | RL | C & D |
| BOHB[†] | 93.61±0.52 | 70.85±1.28 | 44.42±1.49 | 12000 | BO+bandit | C & D |
| ENAS[‡] [4] | 93.76±0.00 | 71.11±0.00 | 41.44±0.00 | 15120 | RL | C |
| DARTS (1st)[‡] [5] | 54.30±0.00 | 15.61±0.00 | 16.32±0.00 | 16281 | gradient | C |
| DARTS (2nd)[‡] [5] | 54.30±0.00 | 15.61±0.00 | 16.32±0.00 | 43277 | gradient | C |
| GDAS[‡] [29] | 93.44±0.06 | 70.61±0.21 | 42.23±0.25 | 8640 | gradient | C |
| DrNAS[♯] [30] | 93.98±0.58 | 72.31±1.70 | 44.02±3.24 | 14887 | gradient | C |
| NASWOT [6] | 92.96±0.81 | 69.98±1.22 | 44.44±2.10 | 306 | training-free | C & D |
| TE-NAS [7] | 93.90±0.47 | 71.24±0.56 | 42.38±0.46 | 1558 | training-free | C |
| KNAS [12] | 93.05 | 68.91 | 34.11 | 4200 | training-free | C & D |
| NASI [8] | 93.55±0.10 | 71.20±0.14 | 44.84±1.41 | 120 | training-free | C |
| GradSign [31] | 93.31±0.47 | 70.33±1.28 | 42.42±2.81 | - | training-free | C & D |
| HNAS ($\mathcal{M}_{\text{Grad}}$) | **94.04**±0.21 | 71.75±1.04 | **45.91**±0.88 | 3010 | hybrid | C & D |
| HNAS ($\mathcal{M}_{\text{SNIP}}$) | 93.94±0.02 | 71.49±0.11 | **46.07**±0.14 | 2976 | hybrid | C & D |
| HNAS ($\mathcal{M}_{\text{GraSP}}$) | **94.13**±0.13 | **72.59**±0.82 | **46.24**±0.38 | 3148 | hybrid | C & D |
| HNAS ($\mathcal{M}_{\text{Trace}}$) | **94.07**±0.10 | **72.30**±0.70 | 45.93±0.37 | 3006 | hybrid | C & D |
| **Optimal** | 94.37 | 73.51 | 47.31 | - | - | - |

[†] Reported by Dong and Yang [23].
[‡] Re-evaluated using the codes provided by Dong and Yang [23].
[♯] Re-evaluated under a comparable search budget as other training-based NAS algorithms with first-order optimization, e.g., ENAS and DARTS (1st). Note that this search budget is smaller than the one reported in its original paper and hence will lead to decreased search performances.

## 6.4 Effectiveness and Efficiency of HNAS

To justify that our theoretically motivated HNAS is able to enjoy the advantages of both training-free (i,e., the superior search efficiency) and training-based (i.e., the remarkable search effectiveness) NAS, we compare it with other baselines in NAS-Bench-201 (Table 3). We refer to Appendix B.2 for our experimental details. As summarized in Table 3, HNAS, surprisingly, advances both training-based and training-free baselines by consistently selecting architectures achieving the best performances, leading to smaller gaps toward the optimal test errors in the search space. Meanwhile, HNAS requires at most 13× lower search costs than training-based NAS algorithms, which is even smaller than the training-free baseline KNAS. Moreover, thanks to the superior evaluation efficiency of training-free metrics, HNAS can be deployed efficiently in not only continuous (where search space is represented as a supernet) but also discrete search space. As for NAS under limited search budgets (Figure 2), HNAS also advances all other baselines by achieving improved search efficiency and effectiveness. Appendix C.5 further includes the impressive search results achieved by HNAS on CIFAR-10/100 and ImageNet in the DARTS search space. Overall, our HNAS is indeed able to enjoy the advantages of both training-free (i.e., the superior search efficiency) and training-based NAS (i.e., the remarkable search effectiveness), which consistently boosts existing training-free NAS methods.

## 7 Conclusion & Discussion

This paper performs a unified theoretical analysis of NAS algorithms using gradient-based training-free metrics, which allows us to *(a)* theoretically unveil the connections among these training-free metrics, *(b)* provide theoretical guarantees for the empirically observed compelling performance of these training-free NAS algorithms, and *(c)* exploit these theoretical understandings to develop a novel framework called HNAS that can consistently boost existing training-free NAS. We expect that our theoretical understanding to provide valuable prior knowledge for the design of training-free

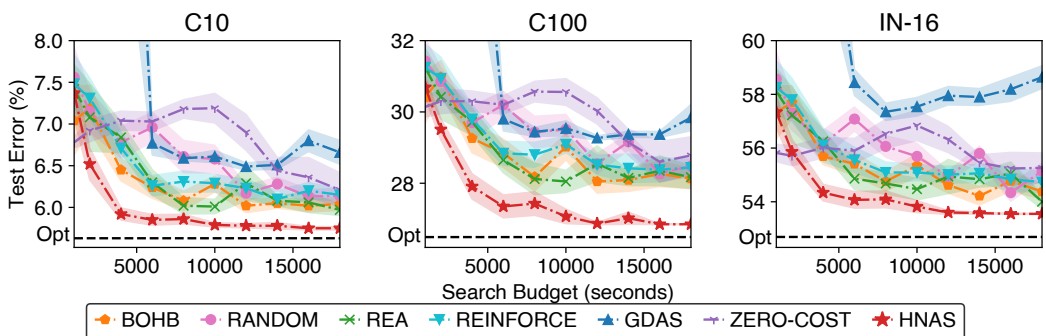

Figure 2: Comparison between HNAS ($\mathcal{M}_{\text{Trace}}$) and other NAS baselines in NAS-Bench-201 under varying search budgets. Here, the ZERO-COST method is borrowed from [9] by using $\mathcal{M}_{\text{Trace}}$. Note that each algorithm is reported with the mean and standard error of ten independent searches, and the black dashed line in each plot denotes the the minimal (optimal) test error that can be achieved by the architectures in NAS-Bench-201 on the corresponding dataset.

metrics and NAS search space in the future. Moreover, we expect our theoretical analyses for DNNs to be capable of inspiring more theoretical understanding and improvement over existing machine learning algorithms that are based on DNNs, e.g., the recent training-free data valuation algorithm [32]. In addition, the impressive performance achieved by our HNAS framework is expected to be able to encourage more attention to the integration of training-free and training-based approaches in other fields in order to enjoy the advantages of these two types of methods simultaneously.

## Acknowledgments and Disclosure of Funding

This research/project is supported by the National Research Foundation Singapore and DSO National Laboratories under the AI Singapore Programme (AISG Award No: AISG2-RP-2020-018) and by A*STAR under its RIE2020 Advanced Manufacturing and Engineering (AME) Programmatic Funds (Award A20H6b0151).

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
