# OpenReview forum: "Unifying and Boosting Gradient-Based Training-Free Neural Architecture Search"
_NeurIPS.cc/2022/Conference — NeurIPS 2022 Accept_

### Official Review · Reviewer_syT9 · 2022-07-09

**Rating:** 6
**Confidence:** 3
**Soundness:** 3 good
**Presentation:** 3 good
**Contribution:** 3 good

**Summary:**

The submission aims to put a stronger theoretical foundation under a set of surrogate metrics that are used in training-free network architecture search and then derive a new method with theoretical and empirical improvement over prior art. "Surrogate"  in this context means "not using data, but only initialized weight parameters", where an increase in the metric is demonstrated empirically to correlate with better model performance. The authors observe that prior art lacks a common explanation of why these metrics work well in practice and they endeavor to provide a common, explanatory model and subsequently a theoretical and empirical improvement motivated from the proposed explanatory model.

The new model essentially comes in two steps: (a) deriving an upper bound for the model generalization error as sum of the training error and a function that is related to the inverse ratio of any such surrogate metric (the latter term being a form of generalization gap and the inverse ratio conforming with the empirical observation that larger surrogate metrics correlate with better model performance), and (b) replacing the training error (first term in the upper bound) by an upper bound as function of only the surrogate metric and the number of samples (but not the samples themselves). These two steps lead to an upper bound of the generalization error that is not a function of the samples themselves, but only the surrogate metric (which is assumed to be a function not of the samples itself) and a set of nuisance parameters (some relating to the initialized model weights). Finally, the paper suggest to aggregate the free nuisance parameters (ones that are not a function of the model/weight) into two generic hyperparameters, reducing the upper bound on the generalization error to a function of only two variables and the model structure and weights. The paper goes on to propose optimizing these hyperparameters via Bayesian optimization using Gaussian processes in an iterative two-step process, where the first step is training-free and optimizes the chosen training free metric over the set of possible networks, and the second step updates the Bayesian optimization with the true validation loss of the chosen network.

The submission exposes several further theoretical connections: (a) demonstrating that one specific metric is an upper bound to the others, (b) relating the condition number of the initial neural tangent kernel to depth and width of networks (under simplifying assumptions), resulting in interpretations that are consistent with earlier empirical results.



**Questions:**

Summarizing the above points that would help clarify the exposure better:

1. How is (7) solved in Algorithm 1?
2. How many iterations K in Algorithm 1 do the presented experiment require? I understand that this is the number of times that an actual training on data is performed in the hybrid setting.
3. Could you please share more details on the BO in section 5.2? The presented material leaves questions open: What are the open parameters in this optimization, how do the choices influence the optimizations? I feel that this part is important to gauge the overall quality and significance of the paper and it is a bit glanced over.

As it is I am rating the paper as borderline accept, primarily as I feel that some of the aspects in the questions (1), and (3) are missing to value the full contribution of the submission.

[Update post-rebuttal] I feel that my questions from the review have been addressed well and I am increasing my soundness score (to "good") as well as the rating (to "weak accept") respectively.

**Limitations:**

The submission highlights limitations sufficiently.

**Strengths And Weaknesses:**

The paper addresses an interesting problem (improved bounds and methods for metrics in training-free architecture search). The authors highlight their contributions and put them in the context of prior work reasonably well. I appreciate the novel combination of training free metrics and training-derived performance, both from a theoretical as well as practical perspective.

It appears that the main contribution of the paper lies in deriving Corollary 2 and subsequently the practical procedure in Sec 5.1. This result appears to generalize several prior art (unified view on training-free metric) as well as bound the generalization performance more tightly than previous models (by making the more realistic assumption of a non-zero achievable training error). To my knowledge this is a substantial, original result and including the subsequent optimization scheme appears to demonstrate favorable results. The empirical results (mainly Table 3) when taken out of context of the theoretical foundation could be seen as not sufficiently convincing: the achieved accuracy is generally slightly better than the compared methods, but the runtime is significantly worse when compared to some. Admittedly, the acceptable tradeoff between run-time and accuracy may be domain-specific ("train fast or well"), thus both priorities have their value. The slight accuracy improvement, most prominent on the IN-16 dataset, is consistent with the presented theoretical perspective and thus good substantiation of the former. From an empirical perspective, it may perhaps be insightful to test the method in a NAS regimen where the current state of the art is still far worse than the true best accuracy to highlight a different, valuable trade off between accuracy and training time.

I had some issues with the clarity and coherence of the paper:
- There are two parts that on first reading appear disjoint with the rest of the paper. First, the relation between the metrics in theorem 1 does not appear to have a bearing on the subsequent deductions. Only after following the proof in appendix A3 the importance becomes clearer. This is a minor point and could be improved by explicitly stating the relation of Theorem 1 to Corollary 2 in the main paper. Secondly, and also very minor by itself, Section 4.4 does not seem to serve a purpose in the main paper except for highlighting speculative connections based on assuming models of simple topologies (deep vs. wide). I feel that the paper would benefit from either omitting or expanding this section.
- More importantly, there appear to be details missing that I would find significant: (a) how is the Bayesian optimization implemented, and (b) how is (7) solved for in Algorithm 1.

Originality: The paper is an extension of previous work that adds original thought and explorations.
Quality: The paper is written well, it puts the presented work in context well and combines theoretic with empirical results.
Significance: The main significance as it is exposed appears to me in Corollary 2 and the subsequent derivations of a practical optimization scheme for training-free/hybrid NAS. The significance of this result would be more empirically evident if the empirical results would show achievable trade-offs between accuracy and training time more differentiated from the state of the art as it does now.

---

> ### Author Response · Authors · 2022-08-01
> **Response #1 to Reviewer syT9**
>
> We appreciate your valuable feedback and constructive comments on improving our paper. We would like to address your concerns below.
>
> ---
> > The significance of this result would be more empirically evident if the empirical results would show achievable trade-offs between accuracy and training time more differentiated from the state of the art as it does now.
>
> We appreciate your valuable feedback on how to further improve the significance of our empirical results! To achieve this, we may need to construct a new NAS benchmark on more complex datasets compared with the relatively "simple" ones (e.g., CIFAR-10/100 and IN-16) such that the current SOTA will be far worse than the true best accuracy in the search space. Existing NAS benchmarks cannot achieve this  because they are mainly constructed on these relatively "simple" datasets and thus, current SOTA is usually close to the optimum in practice. Unfortunately, we currently cannot afford to construct such a new benchmark due to its substantial computational cost, which nevertheless will be an interesting work for us to explore in the future. On the other hand, as we know, it usually becomes harder and more costly to achieve performance improvement in practice when the results are approaching the optimum. In light of this, our empirical results in Table 3 should already be quite significant as it can achieve consistent improvement (i.e., noticeably higher test accuracy on all the three datasets in Table 3) over existing SOTA training-based and training-free NAS algorithms by incurring at most 13$\times$ lower search costs than training-based algorithms.
>
> > There are two parts that on first reading appear disjoint with the rest of the paper...
>
> We thank you for your constructive suggestions on the coherence of this paper! We would like to follow your suggestion to explicitly state the relation of Theorem 1 to Corollary 2 in our main paper. As for  Sec. 4.4, it actually aims to further emphasize the importance of $\kappa$ in Corollary 2 and hence to justify the superiority of Corollary 2 (to Corollary 1) by theoretically connecting it with the topology preference of training-based NAS algorithms. Specifically, we show in Sec. 4.4 that a relatively small $\kappa$ typically corresponds to a wide topology that usually achieves a lower generalization error, as suggested by Corollary 2 (see line 229 in our main paper). This actually aligns with the finding that a wide topology is usually selected by training-based NAS algorithms due to its competitive performance in [19] (see line 230 in our main paper), thus implying that Corollary 2 is more practical and hence superior to Corollary 1, as validated empirically in Table 1 in our main paper. We will include this clarification in our revised paper for its coherence.
>
> > How is (7) solved in Algorithm 1?
>
> To solve Eq. 7 for every step of HNAS, in practice, we independently and randomly sample a large pool (e.g., with a size of 2000 for NAS-Bench-201) of architectures from the search space and then select the architecture achieving the optimum value of Eq. 7 (given the values of $\mu$ and $\nu$) from all sampled architectures (e.g., $2000 \times k$ architectures for step $k$ of our HNAS). This sampling and then optimizing procedure is similar to the one being used in the NASWOT paper.
> Note that following [9], the training-free metrics of these sampled architectures are evaluated using a batch (with size 64) of sampled data (i.e., line 287 and footnote 2).
> Besides, in practice, we have used a 20-step HNAS in our experiments on NAS-Bench-201. So, the number of all sampled architectures is about 40,000, which can already cover all the architectures in NAS-Bench-201 (containing 15,625 architectures) with a high probability. This means that Eq. 7 can be solved efficiently and effectively following our technique above.
>
> > How many iterations K in Algorithm 1 do the presented experiment require?
>
> In practice, we set K to be 20 for NAS-Bench-201, which is shown to be large enough for our Algo. 1 to achieve competitive results.

---

> > ### Author Response · Authors · 2022-08-01
> > **Response #2 to Reviewer syT9**
> >
> > > Could you please share more details on the BO in section 5.2? ... What are the open parameters in this optimization, how do the choices influence the optimizations?
> >
> > BO is a type of gradient-free optimization algorithm aiming to optimize a black-box or non-differentiable objective function by iteratively selecting an input (to only evaluate/query its function value) that intuitively trades off between sampling an input with a likely optimum given the current (possibly imprecise) belief of the function modeled by a Gaussian process (GP) (i.e., exploitation) vs. improving the GP belief of the function over the entire input domain (i.e., exploration) to guarantee finding the global optimum. Since we adopt the non-differentiable validation performance (i.e., validation accuracy) as the objective function to be optimized (over $\mu$ and $\nu$) in our Algo. 1, BO will naturally be a better choice to find the optimal $\mu$ and $\nu$ compared with gradient-based optimization algorithms, and therefore has been used in our HNAS. In particular, in every step $k$ of BO, a GP belief with mean $u(\mu, \nu)$ and variance $\sigma^2(\mu,\nu)$ for the entire input domain is firstly obtained following Eq. 1 in [R1] (i.e., by letting input $x$ in [R1] be the column vector $(\mu, \nu)^{\top}$ and the function value $y$ in [R1] be the validation accuracy $\mathrm{ValAcc}(f_{A^*})$) based on the historical observations $\\{(\mu_i, \nu_i, \mathrm{ValAcc}(f_{A^*_{i}}))\\}\_{i=1}^{k-1}$; this corresponds to line 6 in Algo. 1 for step $k-1$. Then, the mean $u(\mu, \nu)$ and standard deviation $\sigma(\mu,\nu)$ from the resulting GP belief are used to construct an acquisition function such as the expected improvement (EI) from [R2] or the upper confidence bound (UCB) $u(\mu, \nu) + \sqrt{\beta} \sigma(\mu,\nu)$ from [R1] where the parameter $\beta > 0$ is set to trade off between exploitation vs. exploration for guaranteeing no regret asymptotically with high probability. Finally, an input (i.e., $\mu_k,\nu_k$) will be selected (for querying) by maximizing the acquisition function within the entire input domain (i.e., line 3 in Algo. 1), e.g., $(\mu_k,\nu_k)=\arg\max_{(\mu,\nu)} u(\mu, \nu) + \sqrt{\beta} \sigma(\mu,\nu)$ for UCB. Note that the acquisition function in BO is usually differentiable and thus gradient-based optimization algorithms (e.g., L-BFGS and gradient ascent) can be applied to maximize it. We'd like to refer you to [R1] for more technical details about BO based on UCB and [this repo](https://github.com/fmfn/BayesianOptimization) for the implementation of BO that has been used in our experiments.
> >
> > The choice of acquisition function and its hyperparameters (e.g., $\beta$ in UCB) are the main open parameters in BO. By default, we adopt the widely applied expected improvement (EI) with its default hyperparameters in [this repo](https://github.com/fmfn/BayesianOptimization) as the acquisition function for the BO in our experiments. For your concern about how these open parameters will influence the optimization, we mainly compare the results from using different acquisition functions (i.e., expected improvement (EI) vs. upper confidence bound (UCB)) and using HNAS($M_{\text{Trace}}$) and HNAS($M_{\text{Grad}}$) on NAS-Bench-201 in the table below. Since the default hyperparameters for different acquisition functions in [this repo](https://github.com/fmfn/BayesianOptimization) have already been tuned for a variety of tasks, we directly make use of them in our experiments without any changes. Interestingly, the results below show that different acquisition functions typically have limited influence on the BO part of our HNAS. That is, our HNAS is shown to be relatively robust to the change of acquisition function in BO.
> >
> > | Algorithm                       | CIFAR-10 (Acc %) | CIFAR-100 (Acc %) | IN-16 (Acc %) |
> > |:--------------------------------|:----------------:|:-----------------:|:-------------:|
> > | HNAS($M_{\text{Grad}}$) w/ EI  | 94.04            | 71.75             | 45.91         |
> > | HNAS($M_{\text{Grad}}$) w/ UCB | 94.05            | 72.04             | 45.81         |
> > | HNAS($M_{\text{Trace}}$) w/ EI  | 94.07            | 72.30             | 45.93         |
> > | HNAS($M_{\text{Trace}}$) w/ UCB | 94.10            | 72.48             | 46.30         |
> >
> > We will add the above clarifications and results into our revised paper.
> >
> > [R1] Gaussian process optimization in the bandit setting: No regret and experimental design. *Proc. ICML*, 2010.
> >
> > [R2] Efficient global optimization of expensive black-box functions. *J. Global Optimization*, 13:455-492, 1998.
> >
> > ---
> >
> > We sincerely hope our clarifications above have well addressed your concerns, especially for the ones in Q.1 and Q.3, and can improve your opinion of our work.

---

> ### Author Response · Authors · 2022-08-09
> **Thanks to reviewer syT9**
>
> We would like to thank the reviewer again for appreciating the theoretical contributions of our paper and also for the constructive comments to further improve our paper. Kindly let us know if you might have further comments on our paper, and we would like to do our best to address them in the remaining time.

---

### Official Review · Reviewer_VFc4 · 2022-07-11

**Rating:** 6
**Confidence:** 4
**Soundness:** 3 good
**Presentation:** 4 excellent
**Contribution:** 3 good

**Summary:**

This paper aims to understand the training-free NAS metrics from a theoretical perspective and build a unified theoretical framework to analyze the relationships of gradient-based training-free NAS metrics. Based on the theoretical results, the authors further propose a hybrid NAS (HNAS) method which can enjoy the advantages of both training-free (i.e., superior search efficiency) and training-based (i.e., remarkable search effectiveness) NAS. Experimental results demonstrate the effectiveness of the proposed method.

**Questions:**

1. In Table 3, could the authors also provide the comparison of query numbers (query of training-free metrics and training-based metrics) of different methods, rather than only the total search costs of GPU time?. In addition, more SOTA methods on NAS-Bench-201 such as DrNAS (DrNAS: Dirichlet Neural Architecture Search, ICLR 2021) are recommended to be included in Table 3. These help to evaluate the proposed HNAS more thoroughly.

2. Could the authors provide the Spearman/K’Tau results between the unified objective in Equation (7)/(6) and Ground truth accuracies? which helps to further show the superiority of HNAS. If this is suitable for evaluating HNAS?

**Limitations:**

No limitations.

**Strengths And Weaknesses:**

++The proposed HNAS is well-motivated, technical sound, and theoretically guaranteed. The paper is well-written and easy to follow.

--The experimental results are a bit weak since the authors only conduct experiments on NAS-Bench-201. For better evaluation, results on ImageNet would be more convincing.

---

> ### Author Response · Authors · 2022-08-01
> **Response to Reviewer VFc4**
>
> We thank you for taking the time to review our paper and for your valuable feedback. We would like to address your concerns below.
>
> ---
>
> > The experimental results are a bit weak since the authors only conduct experiments on NAS-Bench-201. For better evaluation, results on ImageNet would be more convincing.
>
> In fact, besides the experiments on NAS-Bench-201 in our main paper, we have also conducted experiments on CIFAR-10/100 and ImageNet using the DARTS search space. The results have already been summarized in our Appendix C.4, which can further demonstrate the superiority of our HNAS over other training-free (with improved search effectiveness) and training-based NAS algorithms (with reduced search costs).
>
> > In Table 3, could the authors also provide the comparison of query numbers (query of training-free metrics and training-based metrics) of different methods, rather than only the total search costs of GPU time? In addition, more SOTA methods on NASBench-201 such as DrNAS are recommended to be included in Table 3.
>
> Due to the methodological diversity of recent NAS algorithms, it may not be meaningful to provide a comparison about the query number of different methods in Table 3 in our main paper. To clarify this, on the one hand, most recent training-based NAS algorithms (e.g., ENAS, DARTS, and DrNAS) are based on the one-shot model training of a supernet. As a result, the query number of training-based or training-free metrics cannot be defined meaningfully in these algorithms. On the other hand, the training cost of different architectures (e.g., in NAS-Bench-201) usually varies. So, the query number itself may not be good enough to provide a fair comparison about the efficiency of different query-based NAS algorithms (e.g., BOHB and RS). So, we directly follow the *convention in the NAS literature* to utilize the search cost (measured as GPU seconds) for comparing the performances of different NAS algorithms in our main paper.
>
> We thank you for pointing out the DrNAS baseline. We follow your suggestion to compare its performance (using a search budget of $<$15,000 GPU seconds) with that of our HNAS ($M_{\text{GraSP}}$) on NAS-Bench-201 and report the corresponding results in the table below, which will be included in our revised paper. Note that we have applied a smaller search budget for DrNAS compared with the one in its original paper (about 24,000 GPU seconds on a Nvidia 1080Ti) because we intend to maintain comparable search budgets for different training-based NAS algorithms using first-order optimization (e.g., ENAS, GDAS, DARTS (1st)). The results show that our HNAS can also achieve better search performances than DrNAS (with a search budget of 14887 GPU Seconds) when incurring about 4$\times$ lower search cost.
>
> | Algorithm                 | CIFAR-10 (Acc %) | CIFAR-100 (Acc %) | IN-16 (Acc %) | Cost (GPU Sec.) |
> |:--------------------------|:--------:|:---------:|:-----:|:----:|
> | DrNAS                  | 93.98    | 72.31     | 44.02 | 14887 |
> | HNAS ($M_{\text{GraSP}}$) | **94.13**    | **72.59**     | **46.24** | **3148** |
>
> > Could the authors provide the Spearman/K’Tau results between the unified objective in Equation (7)/(6) and Ground truth accuracies? which helps to further show the superiority of HNAS. If this is suitable for evaluating HNAS?
>
> In fact, the Spearman/K’Tau results between the unified objective in Eq. 7 and ground truth test errors have already been summarized in the non-realizable scenario in Table 1 in our main paper, which are shown to be better than those using training-free metrics directly (i.e., realizable scenario in Table 1) and can thus imply the superiority of our HNAS via Eq. 7.
>
> ---
>
> We sincerely hope our clarifications above have addressed your concerns and can improve your opinion of our work.

---

> > ### Comment · Reviewer_VFc4 · 2022-08-08
> > **Thank you for your response**
> >
> > Thank you for your response and most of my concerns have been addressed. I apologize that I did not observe your ImageNet results in Appendix. As for the query number, I still think it is also a good metric to evaluate the efficiency of a pure ‘Search Strategy’ (do not consist of the cost of an architecture estimator, such as a super network). However, this is not a big issue and I believe HNAS has lower query numbers since query number is also somehow in proportion to search costs. Thus, I will increase my rating.

---

> > > ### Author Response · Authors · 2022-08-09
> > > **Thank you for raising your score!**
> > >
> > > We are glad to know that our response has addressed most of your concerns with clarity. We would also like to thank you for raising your score for our paper. As for the query number of only pure ‘Search Strategy’ (i.e., REA, RS, and REINFORCE in Table 3 in our main paper) that you are interested in, we have further provided the comparison in the following table, which will also be included in our revised paper. As you have mentioned, the query number is roughly in proportion to the search cost, and our HNAS indeed incurs a smaller query number to achieve better performance.
> > >
> > > | Algorithm                 | CIFAR-10 (Acc %) | CIFAR-100 (Acc %) | IN-16 (Acc %) | Cost (GPU Sec.) | Numer of Queries |
> > > |:--------------------------|:--------:|:---------:|:-----:|:----:|:----:|
> > > | REA                 |  93.92    | 71.84     | 45.15 | 12000 | 102 |
> > > | RS (w/o sharing)                  |  93.70    | 71.04     | 44.57 | 12000 | 106 |
> > > | REINFORCE                 |  93.85    | 71.71    | 45.24 | 12000 | 103 |
> > > | HNAS ($M_{\text{GraSP}}$) | **94.13**    | **72.59**     | **46.24** | **3148** | **20** |

---

### Official Review · Reviewer_cbB5 · 2022-07-11

**Rating:** 6
**Confidence:** 3
**Soundness:** 3 good
**Presentation:** 3 good
**Contribution:** 3 good

**Summary:**

This paper theoretically derives the connections among existing gradient-based metrics for training-free neural architecture search, and then develops a generalization bound based on these metrics, which explains why these metrics are correlated with model performance in a rigorous way. Two cases (zero and non-zero training error) are further analyzed. Based on the theoretical results, a unified objective and its corresponding BO-based search algorithm (HNAS) are proposed. Experiments on popular NAS benchmarks are conducted to show the validity of the theoretical results and the proposed algorithm.

**Questions:**

See the weakness concerns listed in the last section.

**Limitations:**

I do not see an explicit discussion about the limitations of the method in this paper.

There is no concern about negative societal impact.

**Strengths And Weaknesses:**

This paper is well organized and has a clear presentation.
The theoretical perspective is very interesting. The generalization bound offers a unified view to inspect and interpret current training-free NAS metrics. I mainly have the following concerns:

1.	The term O(k/M) plays the role of generalization gap. In M_{trace}, the trace norm of neural tangent kernel is highly correlated with gradient norm. The eigenvalues of neural tangent kernel matrix also affect gradient norm. So, are k and M correlated? How to ensure that decreasing k while increasing M would not cause optimization conflict?

2.	The details of the proposed algorithm, HNAS, are not clear enough. How to calculate neural tangent kernel matrix and its eigenvalues efficiently has been a problem due to the large parameter dimension and the large number of samples in a dataset. So, the detailed process of solving Eq. (7) should be described. Besides, the BO step to optimize \mu and \nu is also unclear. Is the validation performance differentiable to BO? Detailed description should be offered.

3.	The authors should compare their proposed NAS algorithm with simply using the metrics. For example, replace Eq. (7) with \max M_{grad}, M_{snip}, M_{grasp}, or M_{Trace}, to show the effectiveness of the upper bound in Corollary 2, and the importance of BO. Besides, more recent NAS methods, such as [1], should be compared with.

[1] Zhang et al., GradSign: Model Performance Inference with Theoretical Insights, ICLR 2022.

---

> ### Author Response · Authors · 2022-08-01
> **Response #1 to Reviewer cbB5**
>
> We deeply appreciate your valuable feedback and constructive comments on improving our paper. We would like to address your questions below.
>
> ---
>
> > So, are k and M correlated? How to ensure that decreasing k while increasing M would not cause optimization conflict?
>
> Since condition number $\kappa$ is defined as the ratio of the maximum and minimum eigenvalues of the NTK matrix and $M_{\text{Trace}}$ can be upper-bounded by the maximum eigenvalue of NTK, $\kappa$ and $M_{\text{Trace}}$ would usually be related in practice, which is also supported by the results in Table 2 in our main paper, i.e.,
> a wide architecture with a smaller $\kappa$ also yields a smaller $M_{\text{Trace}}$ in general compared with a deep architecture.
> So, there exists a trade-off when optimizing Eq. 7 in our main paper; to be more precise, we refer to this phenomenon as a trade-off rather than an optimization conflict. In fact, both the theoretical analysis in Corollary 2 and the empirical studies in Table 5 show that maintaining this trade-off (i.e., optimizing $\kappa/M$ as a whole) is important for training-free NAS to better characterize the performances of architectures. So, our HNAS optimizes $\kappa/M$ as a whole in Eq. 7 so as to find architectures with a better-performing trade-off, which has shown to be able to achieve better performance than those without considering this trade-off (e.g., NASI and KNAS).
>
> > The details of the proposed algorithm, HNAS, are not clear enough.
>
> We thank you for pointing this out. We will clarify the details of our HNAS one by one below:
>
> 1. In practice, we follow the convention of NASI and TE-NAS to evaluate the NTK matrix (based on the definition in Eq. 1) and its eigenvalues using a batch (with size 64) of randomly sampled data from a dataset, which can finally be evaluated efficiently (e.g., it usually only costs a few seconds in a single GPU).
> 2. To solve Eq. 7 for every step of HNAS, we independently and randomly sample a large pool (e.g., with a size of 2000 for NAS-Bench-201) of architectures from the search space and then select the architecture achieving the optimum value of Eq. 7 (given the values of $\mu$ and $\nu$) from all sampled architectures (e.g., $2000 \times k$ architectures for step $k$ of our HNAS). This sampling and then optimizing procedure is similar to the one being used in the NASWOT paper. Besides, in practice, we have used a 20-step HNAS in our experiments on NAS-Bench-201. So, the number of all sampled architectures is about 40,000, which can already cover all the architectures in NAS-Bench-201 (containing 15,625 architectures) with a high probability. This means that Eq. 7 can be solved efficiently and effectively following our technique above.

---

> > ### Author Response · Authors · 2022-08-01
> > **Response #2 to Reviewer cbB5**
> >
> > 3. BO is a type of gradient-free optimization algorithm aiming to optimize a black-box or non-differentiable objective function by iteratively selecting an input (to only evaluate/query its function value) that intuitively trades off between sampling an input with a likely optimum given the current (possibly imprecise) belief of the function modeled by a Gaussian process (GP) (i.e., exploitation) vs. improving the GP belief of the function over the entire input domain (i.e., exploration) to guarantee finding the global optimum. Since we adopt the non-differentiable validation performance (i.e., validation accuracy) as the objective function to be optimized (over $\mu$ and $\nu$) in our Algo. 1, BO will naturally be a better choice to find the optimal $\mu$ and $\nu$ compared with gradient-based optimization algorithms, and therefore has been used in our HNAS. In particular, in every step $k$ of BO, a GP belief with mean $u(\mu, \nu)$ and variance $\sigma^2(\mu,\nu)$ for the entire input domain is firstly obtained following Eq. 1 in [R1] (i.e., by letting input $x$ in [R1] be the column vector $(\mu, \nu)^{\top}$ and the function value $y$ in [R1] be validation accuracy $\mathrm{ValAcc}(f_{A^*})$) based on the historical observations $\\{(\mu_{i}, \nu_{i}, \mathrm{ValAcc}(f_{A^*_{i}}))\\}\_{i=1}^{k-1}$; this corresponds to line 6 in Algo. 1 for step $k-1$. Then, the mean $u(\mu, \nu)$ and standard deviation $\sigma(\mu,\nu)$ from the resulting GP belief are used to construct an acquisition function such as the expected improvement (EI) from [R2] or the upper confidence bound (UCB) $u(\mu, \nu) + \sqrt{\beta} \sigma(\mu,\nu)$ from [R1] where the parameter $\beta > 0$ is set to trade off between exploitation vs. exploration for guaranteeing no regret asymptotically with high probability. Finally, an input (i.e., $\mu_{k},\nu_{k}$) will be selected (for querying) by maximizing the acquisition function within the entire input domain (i.e., line 3 in Algo. 1), e.g., $(\mu_k,\nu_k)=\arg\max_{(\mu,\nu)} u(\mu, \nu) + \sqrt{\beta} \sigma(\mu,\nu)$ for UCB. Note that the acquisition function in BO is usually differentiable and thus gradient-based optimization algorithms (e.g., L-BFGS and gradient ascent) can be applied to maximize it. We'd like to refer you to [R1] for more technical details about BO based on UCB and [this repo](https://github.com/fmfn/BayesianOptimization) for the implementation of BO that has been used in our experiments.
> >
> > We will add the above details into our revised paper for completeness.
> >
> > [R1] Gaussian process optimization in the bandit setting: No regret and experimental design. *Proc. ICML*, 2010.
> >
> > [R2] Efficient global optimization of expensive black-box functions. *J. Global Optimization*, 13:455-492, 1998.
> >
> > > The authors should compare their proposed NAS algorithm with simply using the metrics ... Besides, more recent NAS methods, such as [1], should be compared with.
> >
> > In fact, we have already included the comparison between our HNAS and those simply using the training-free metrics on both (1) the correlation with the test performance of architectures in Table 1 in our main paper (i.e., non-realizable scenario (obtained following HNAS) vs. realizable scenario (from those simply using the training-free metrics)) and (2) the final search results on NAS-Bench-201 in  Table 3 in our main paper (i.e., HNAS vs. any other training-free method). These results indeed help to substantiate the effectiveness of the upper bound in our Corollary 2 and the importance of BO in our HNAS. As for the GradSign paper you have mentioned, we follow your suggestion to compare its performance with that of our HNAS ($M_{\text{GraSP}}$) on NAS-Bench-201 and report the corresponding results in the table below, which will be included in our revised paper. The results show that our HNAS can also achieve better search performances than GradSign.
> >
> > | Algorithm                 | CIFAR-10 (Acc %) | CIFAR-100 (Acc %) | IN-16 (Acc %) |
> > |:--------------------------|:--------:|:---------:|:-----:|
> > | GradSign                  | 93.31    | 70.33     | 43.95 |
> > | HNAS ($M_{\text{GraSP}}$) | **94.13**    | **72.59**     | **46.24** |
> >
> > ---
> >
> > We sincerely hope our clarifications above have addressed your questions and can improve your opinion of our work.

---

> ### Author Response · Authors · 2022-08-09
> **Thanks to reviewer cbB5**
>
> We would like to thank the reviewer again for recognizing the theoretical and empirical contributions of our paper and also for the valuable feedback. Kindly let us know if you might have further comments, and we will do our best to address them in the remaining time.

---

> > ### Comment · Reviewer_cbB5 · 2022-08-09
> > **thanks for the response**
> >
> > Thanks for the response. I suggest that the authors add the discussions of these questions into the revised paper. I would like to keep my original score.

---

### Official Review · Reviewer_TdLx · 2022-07-15

**Rating:** 8
**Confidence:** 4
**Soundness:** 3 good
**Presentation:** 4 excellent
**Contribution:** 4 excellent

**Summary:**

This paper came up with the lower bound of generalization error for several gradient based training-free nas methods. They were able to relate training-free metrics such as $M_{Grad}$, $M_{SNIP}$,  $M_{Grasp}$ with the trace of the the NTK-Matrix. The generalization bound for the scenarios where training error is not zero is proportional k/M where $k$ is the ratio of the largest and the smallest eigen values of the NTK matrik and $M$ is the training-free metric. So they came up with a formulation for generalization error based on $k$ and $M$ involving hyperparameters $\mu$ and $v$

Further they devised a BO-based NAS algorithm, that searches the hyper parameters for the above formulation. The architecture minimizing the generalization error formulation is found and is trained to obtain its validation performance. The BO's surrogate model is then updated with the validation performance. This is continued and the architecture with the best validation accuracy is returned

**Questions:**

Instead of training-free and training based hybrid NAS, can we come up with a completely training-free algorithm based on what we know so far? for example what if we just use k/$M_{Trace}$ as the metric?

**Strengths And Weaknesses:**

Strengths
This is the first paper that is able to relate several gradient based training-free metrics and formulate a generalization bound.
The experiments also show that $M_{Grad}$, $M_{SNIP}$ and  $M_{Grasp}$ are highly correlated with $M_{Trace}$
Their search algorithm is sample-efficient and is only searching for 2 hyper parameters rather than the entire architecture.

---

> ### Author Response · Authors · 2022-08-01
> **Response to Reviewer TdLx**
>
> We thank you for taking the time to review our paper and for your positive feedback. We will answer your question below.
>
> ---
>
> > Instead of training-free and training-based hybrid NAS, can we come up with a completely training-free algorithm based on what we know so far? for example what if we just use $\kappa/M_{\text{Trace}}$ as the metric?
>
> We thank you for this interesting question! A short answer to your first question is yes. According to  Eq. 7 in our main paper, a completely training-free metric can be produced by simply specifying the values of $\mu$ and $\nu$ with prior knowledge. For example, the $\kappa/M_{\text{Trace}}$ you have mentioned is exactly a special case of Eq. 7 with $\mu=0$. However, obtaining prior knowledge regarding the best choice of $\mu$ and $\nu$ for NAS is non-trivial. Therefore, tuning $\mu$ and $\nu$ would be a better alternative to achieve more competitive search results in practice. For example, Fig. 5 in our appendix has empirically shown that tuning $\mu$ and $\nu$ can produce a training-free metric that can better characterize the performances of architectures. To further validate the importance of tuning $\mu$ and $\nu$ for NAS, we also show how the generalization performance (i.e., test accuracy) of the selected architecture evolves during the optimization of our HNAS using $M_{\text{Trace}}$ on NAS-Bench-201 in the table below. The result of step $k=1$ (i.e., first row) in the table below comes from the training-free metric $\kappa/M_{\text{Trace}}$ you have mentioned. Notably, the results in the table show that tuning $\mu$ and $\nu$ based on training-based performance can also lead to improved search results and therefore will be a better alternative than pre-defining $\mu$ and $\nu$ for a completely training-free NAS, which further justifies the essence of combining training-free and training-based methods (as one of our major contributions) in HNAS. We will include these new results and discussion in our revised paper.
>
>
> | Step $k$ | CIFAR-10 (Best Acc %) | CIFAR-100 (Best Acc %) | IN-16 (Best Acc %) |
> |----------|:---------------------:|:----------------------:|:------------------:|
> | 1        | 93.50                 | 69.78                  | 43.73              |
> | 3        | 93.50                 | 69.78                  | 43.73              |
> | 5        | 93.50                 | 69.78                  | 43.73              |
> | 7        | 93.93                 | 71.44                  | 46.13              |
> | 9        | 94.08                 | 72.01                  | 46.13              |
> | 11       | 94.08                 | 72.01                  | 46.13              |
> | ...      | ...                   | ...                    | ...                |
>
> ---
>
> We thank you for appreciating our contributions. We sincerely hope our clarifications above have addressed your question.

---

> > ### Comment · Reviewer_TdLx · 2022-08-07
> > **Thank you for your response**
> >
> > Thanks for answering my question. This table is very insightful. K=1 is not too bad if you want to choose training-free, however searching for the hyperparameters definitely helps.
> > I would like to keep my score

---

> > > ### Author Response · Authors · 2022-08-09
> > > **Thank you for appreciating our response!**
> > >
> > > We are glad to know that our response has addressed your question.

---

### Meta-Review · Area_Chair_Ay62 · 2022-08-26

**Recommendation:** Accept
**Confidence:** Certain

**Metareview:**

The paper is the first to theoretically analyze gradient-based training-free NAS, and come up with a generalization bound. The authors further present a hybrid unified framework to benefit from both training-free and training-based algorithms. The paper is clearly written and technically sound. The reviewers raised several questions about the details of the algorithm and experiments, and the authors addressed them in the response precisely. It is suggested that the authors include those discussions in the revision.

**Award:**

No

---

### Decision · Program_Chairs · 2022-09-14

Accept